# Assessing pyrethroid resistance status in the *Culex pipiens* complex (Diptera: Culicidae) from the northwest suburbs of Chicago, Illinois using Cox regression of bottle bioassays and other detection tools

**Edwin R. Burgess, IV**[1]*, **Kristina Lopez**[2], **Patrick Irwin**[2], **Collin P. Jaeger**[3], **Alden S. Estep**[4]*

**1** Entomology and Nematology Department, University of Florida, Gainesville, Florida, United States of America, **2** Department of Entomology, University of Wisconsin-Madison, Madison, Wisconsin, United States of America, **3** McHenry County College, Crystal Lake, Illinois, United States of America, **4** USDA-ARS-Center for Medical, Agricultural and Veterinary Entomology, Gainesville, Florida, United States of America

* edwinburgess@ufl.edu (ERB); alden.estep@usda.gov (ASE)

## Abstract

*Culex pipiens* complex is an important vector of epizootic and zoonotic pathogens, including West Nile virus. Chicago, Illinois and its suburbs have suffered high incidence of human West Nile virus infections in the past. This makes abatement programs in and around the Chicago area an essential service. The control of *Cx. pipiens* is often complicated by rapidly evolving resistance to pyrethroids, which are the most widely used chemical class in US mosquito abatement programs. The present study assessed Sumithrin® resistance in *Cx. pipiens* collected from five locations around Cook County, Illinois, neighboring the city limits of Chicago. According to CDC guidelines, samples from all five locations demonstrated some resistance to Sumithrin®. When assessed with Anvil®, a formulated product made of Sumithrin® synergized with piperonyl butoxide, susceptibility was rescued in mosquitoes from three out of the five locations, suggesting involvement of mixed-function oxidases and/or carboxylesterases in Sumithrin® resistance at these locations. Not all locations had susceptibility rescued by Anvil®, but these locations had relatively low knockdown resistance allele frequencies, suggesting that mechanisms other than knockdown resistance may be involved. Enzyme activities did not reveal any marked trends that could be related back to mortality in the bottle bioassays, which highlights the need for multiple types of assays to infer enzymatic involvement in resistance. Future directions in pyrethroid resistance management in Chicago area *Cx. pipiens* are discussed.

**Data Availability Statement:** All relevant data are within the paper and its Supporting Information files.

**Funding:** This manuscript was supported in part by Cooperative Agreement Number U01CK000505, funded by the Centers for Disease Control and Prevention to PI and KL. The funders had no role in study design, data collection and analysis, decision to publish, or preparation of the manuscript.

**Competing interests:** The authors have declared that no competing interests exist.

## Introduction

The *Culex pipiens* complex (*Cx. pipiens*) plays a major role in vectoring several epizootic and zoonotic pathogens significant to birds and humans, including West Nile virus (WNV) and St. Louis encephalitis [1, 2]. WNV is considered the most widespread arbovirus in the United States [3]. In the summer of 2002, cases of WNV in Chicago surged, with 884 cases and 66 deaths, followed by another outbreak in 2003 but with only 53 cases [4]. Chicago resides in Cook County, Illinois, and is home to an estimated 5.1 million people. Factors related to geography, housing, population, and abatement strategies around Chicago and the surrounding county makes this a potential high-risk region for WNV cases [5].

Typically thought to feed primarily on birds, *Cx. pipiens* from the upper Midwest are aggressively anthropophagic [6]. The *Cx. pipiens* complex is currently comprised of *Cx. australicus*, *Cx. pipiens*, and *Cx. quinquefasciatus*, with various hybridizations and biotypes among the species, two of which are known as form *pallens* and *molestus* [7]. Form *molestus* is present in Chicago [8] but does not appear to be significantly introgressed into these *Cx. pipiens* populations [9]. The *molestus* form is important because it appears to be more strongly associated with anthropophagy, increasing its potential to transmit pathogens such as WNV [10, 11]. Mosquitoes from the *Cx. pipiens* complex are primarily active at night, which makes them particularly good targets for nightly insecticidal control efforts using ultra-low volume (ULV) fogging [12, 13]. ULV fogging remains the most competent tool available for quickly reducing WNV and other mosquito vector populations during times of arboviral disease outbreaks [14].

ULV products typically contain one of two classes of insecticides, either pyrethroids or rarely organophosphates. Pyrethroids are generally safer than organophosphates to vertebrates, as they break down into safe metabolites relatively quickly and they are cheap. But resistance to pyrethroids has been well-documented in *Cx. pipiens* throughout the United States (reviewed in [15]). Among the likeliest of the proposed physiological mechanisms that confer this resistance are target site mutations in the para region of the voltage-sensitive sodium channel and metabolic detoxification by a few families of enzymes, including mixed-function oxidases (MFO) known as cytochrome P450s, carboxylesterases (CarE), and glutathione *S*-transferases (GST). How these mechanisms combine to confer resistance is not well-understood but nevertheless should be monitored by mosquito control operations to better understand their relationship to product failure.

In the present study, CDC bottle bioassays were conducted as an initial screening for resistance to Sumithrin® (d-phenothrin), a Type I pyrethroid and Anvil® (Sumithrin® plus piperonyl butoxide, a synergist) in *Cx. pipiens* from four sites in the northwest suburbs of Cook County, Illinois. These sites were in Wheeling, Arlington Heights, and two sites in Des Plaines. As a post hoc laboratory analysis, organisms from the bottle bioassays were genotyped for single nucleotide polymorphisms in the voltage-sensitive sodium channel that confers knockdown resistance (*kdr*). Mosquitoes from these samples also were tested for enzymatic activity. Finally, bottle bioassay data was analyzed using clustered Cox regression using time-dependent covariates to account for non-proportional hazards when they occurred [16]. This approach to bottle bioassay data analysis provides additional information on the rate of mortality between start and endpoint. Using Cox regression in this way also is beneficial because non-proportional hazards due to heterogeneity of resistance factors in recently field-derived strains can be accounted for in this type of model.

## Methods

### Mosquito sources and pyrethroid exposure histories

Five study sites were selected within the Northwest Mosquito Abatement District, located in Cook County, IL, USA (Fig 1 and Table 1). Each site is approximately 2.59 km$^2$ and located 0.8

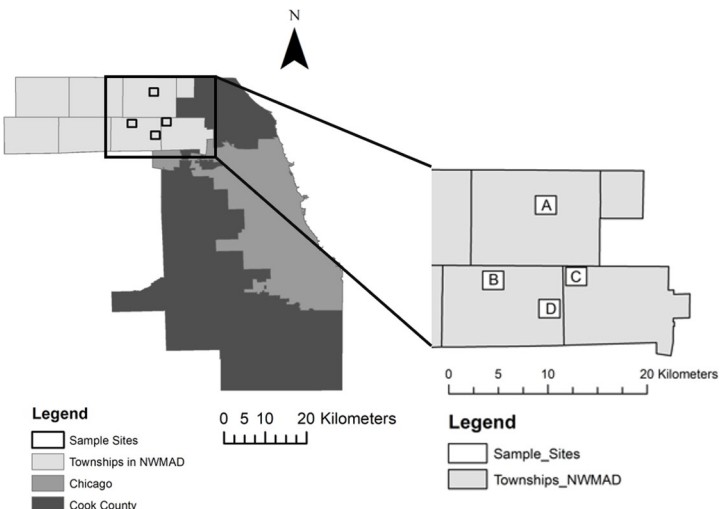

**Fig 1. Trapping locations of the seven strains tested.** A = WHE (Wheeling), B = AHB (Arlington Heights South Catch Basin), C = DPN (Des Plaines North), D = DPS and DPSB (Des Plaines South and Des Plaines South Catch Basin).

km to 8 km from other sites. Within each site, four homeowners allowed mosquito collections with one CDC gravid trap (John W. Hock Company, Gainesville, FL). Gravid traps were baited with an infusion of 75 g Timothy Complete rabbit food (Kaytee Products Inc., Chilton, WI), 58 g Kentucky bluegrass (*Poa pratensis*) collected from the district's property, 2.5 g lactalbumin whey protein (TGS Nutrition, Las Vegas, NV), 0.22 g Altosid® pellets (Central Life Sciences, Schaumburg, IL), and 9.46 L water. Gravid infusion was stored outside for at least one week before use. *Cx. pipiens* egg rafts were collected from gravid trap basins every other week

**Table 1. Sampling collection and analysis information.**

| Strain Location (Latitude, Longitude) | Abbreviation | Date of Bottle Bioassay (Insecticide Used) | Post Hoc Assays[2] |
|---|---|---|---|
| Arlington Heights catch Basin (42.054977,– 87.984398) | AHB | August 5 (Sumithrin®) | Genotype, Enzyme |
| | | August 7 (Anvil®) | None |
| Wheeling (42.117205,– 87.936830) | WHE | August 11 (Sumithrin®) | Genotype, Enzyme |
| | | August 13 (AnvilTM) | None |
| | | August 14 (Anvil®) | None |
| Des Plaines South (42.029653,– 87.930887) | DPS | August 26 (Sumithrin®) | Genotype, Enzyme |
| Des Plaines South catch basin (42.024806,– 87.925787) | DPSB | August 19 (Sumithrin®) | Genotype |
| | | August 21 (AnvilTM, Anvil®) | None |
| Des Plaines North (42.059309,– 87.912542) | DPN[1] | August 5 (Sumithrin®) | None |
| | | August 7 (Anvil®) | Genotype |
| Des Plaines North, first resample | DPN1[1] | August 19 (Sumithrin®) | Enzyme |
| | | August 21 (AnvilTM, Anvil®) | None |
| Des Plaines North, second resample | DPN2[1] | September 1 (Sumithrin®) | Enzyme |

[1] DPN, DPN1, and DPN2 represent different generations of mosquitoes reared from the same sight. DPN2 was collected after DPN1, and DPN1 was collected after DPN.

[2] Genotyping was for the L982F *kdr* mutation (Genotype) using mosquitoes from the treatment bottles, and enzyme quantity and activities tested were mixed-function oxidase, α- and β-carboxylesterase, and glutathione *S*-transferase enzyme activity (Enzyme) using non-treated control mosquitoes run alongside the treated bottles during the bottle bioassays.

between the months of July and August in 2020. To ensure the correct species was collected, egg rafts were hatched and reared individually until species could be identified at second instar [17]. Mosquitoes from the same site and collection day were pooled together and reared at 27˚C and in a 16:8 hour light:dark cycle. Larvae were fed with ground TetraMin® tropical fish flakes (Spectrum Pet Brands LLC, Blacksburg, VA) and adults were fed a 10% sucrose solution.

Sprays were conducted once a week for 5 weeks in Wheeling, Arlington Heights North, and Des Plaines South in 2019 and 2020, starting in July and ending in August. In 2019, Zenivex® E20 (20% etofenprox) in a 1:1 mix with mineral oil (10% etofenprox) was used and in 2020, Anvil® 10+10 was sprayed at 0.0036 Lb per acre of active ingredient and piperonyl butoxide (PBO). From 2013–2018 Northwest Mosquito Abatement District averaged 1 spray event (etofenprox) per year in these areas. The sites where the mosquitoes were obtained are in residential neighborhoods with numerous parks, schools, and manicured lawns. Numerous residential mosquito/pest control companies operate in these residential areas. Pyrethroids are commonly used by homeowners, park districts, schools, and golf courses [18].

## Bottle bioassays

CDC bottle bioassays were completed using three different insecticide solutions on adult *Cx. pipiens* aged 3–6 days [19] (Table 1). Most bottle bioassays were completed with technical grade Sumithrin® (provided in the CDC bottle bioassay kit) diluted with acetone to 20 μg/bottle, the diagnostic dose used in the bottle bioassay guidelines. The diagnostic time for Sumithrin® in *Cx. pipiens* adults is 30 minutes. This time and concentration represent empirically determined parameters that are specific for this species to display 100% mortality when fully susceptible ([19], section 3.2). The second solution consisted of Anvil® 10+10 (Clarke® Mosquito Control Products, Inc., Roselle, IL) diluted to 22.2 μg Sumithrin® and 22.2 μg PBO per bottle [20]. This dose of Sumithrin® is similar to the diagnostic dose in the guidelines. The second Anvil® 10+10 solution was made with the same protocol, instead using the 1:1 mineral oil tank mix (abbreviated AnvilTM) diluted with acetone to 11.1 μg Sumithrin® and 11.1 μg PBO per bottle. 250 mL glass Wheaton bottles were treated with 1 mL insecticide solution according to the CDC procedure and allowed to dry for at least four hours. Insecticide solutions were stored at 4˚C. Control bottles were treated with 1 mL acetone and allowed to dry for four hours. 15–25 mixed sex *Cx. pipiens* were aspirated into each bottle and knockdown was recorded every 5 min for 45 min, then every 15 min until 120 min total. At the completion of the bioassay, mosquitoes were killed and stored at –80˚C.

## Genotyping for knockdown resistance alleles

Genotyping for the 982L and 982F alleles, canonically known as 1014L and 1014F alleles, was conducted using a PCR-based melt curve assay modelled on the assay of Saavedra-Rodriquez et al. [21] but with primers designed for the *Cx. pipiens* complex (Table 2). Controls for the LL, FF, and the LF genotypes, from mosquitoes that had been previously genotyped by Sanger sequencing, were included to ensure that the assay reliably detected all three genotypes common to the US. No template (negative) controls were also included. Individual organisms collected from the CDC bottle bioassays were loaded into 96-well plates (Omni International, Kennesaw, GA) with 400 μL of nuclease free water and cubic zirconium beads (BioSpec Products, Bartlesville, OK). Plates were sealed with Teflon™ sealing mats and homogenized for 60 seconds at 30 hertz (Omni International, Kennesaw, GA). Samples were centrifuged for 1 min at 805g and then maintained on ice until assay setup. PCR master mix was prepared in sufficient quantity for 400 10 μL reactions (2,000 μL SYBR Select, 1,161.2 μL nuclease free water,

**Table 2. Primers and genomic locations for *Cx. pipiens kdr* 1014 melt curve assay.**

| Primer | Genomic location | Sequence |
|---|---|---|
| Cxq_1014L | NC51862.1: 22646376–22646398 | **GCGGGCAGGGCGGCGGGGGCGGGG**TTCACGCTGGAATACTCACGACTA |
| Cxq_1014F | NC51862.1: 22646376–22646397 | **GG**TTCACGCTGGAATACTCACGACA |
| Cxq_1014S | NC51862.1: 22646376–22646398 | **AGCGCGGAGCGCGG**TTCACGCTGGAATACTCACGACTG |
| Cxq_1014_3' | NC51862.1: 22646459–22646483 | GGATCGAATCCATGTGGGACTGCAT |

Primer sequences in bold are tails added to change melting temperatures following the method of [21].

2.4 μL of Cxq_1014L primer, 13.2 μL of Cxq_1014F primer, 10.0 μL of Cxq_1014S primer, and 13.2 μL of Cxq_1014_3' primer). Eight microliters of mastermix was added to each well of a 384-well plate (Thermo Fisher Scientific, Waltham, MA) followed by 2 μL of centrifuged homogenate using an epMotion 5075 liquid handling system (Eppendorf, Hamburg, Germany) with filtered tips. Amplification and melt curve data were collected on an Applied Biosystems QS6 (Thermo Fisher Scientific, Waltham, MA) using default fast cycling conditions. Determination of alleles present in a sample was assessed by examination of the derivative melt curve for temperature peaks ($T_m$) as in [21] (Fig 2). In this assay, an LL genotype is characterized by a distinct $T_m$ of 85.3 ± 0.5°C, an FF by a $T_m$ of 82.6 ± 0.5°C, and an LF heterozygote has peaks at both $T_m$s (Fig 2). Samples that did not amplify or that amplified with a cycle threshold greater than 35 were excluded from analysis.

## Enzyme activity assays

The content of mixed-function oxidases (MFO), and the activity of α- and β-carboxylesterases (α-, β-CarE), and glutathione *S*-transferases (GST) were determined using modified methods

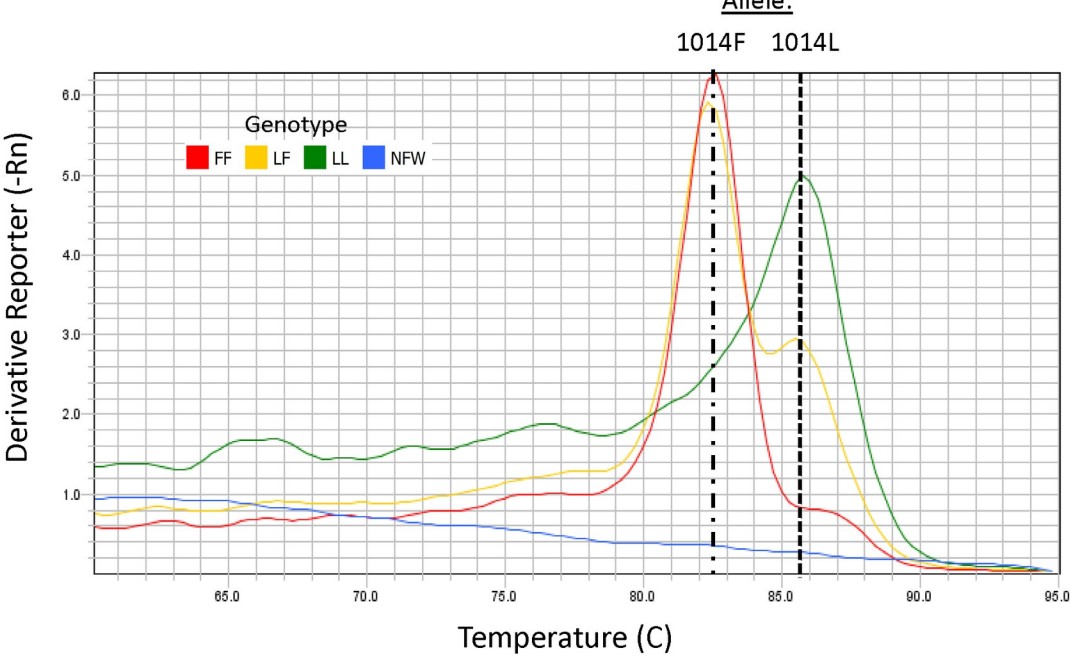

**Fig 2. *Cx. pipiens kdr* 1014 melt curve assay analysis.** Representative melt curve assay results for a 1014L homozygote (LL), a 1014F homozygote (FF), a 1014LF heterozygote (LF), and a nuclease-free water blank (NFW) showing distinct melting temperatures for the L (85.3°C) and F (82.6°C) alleles.

from [22] using only freeze-killed control mosquitoes from the bottle bioassays (i.e., mosquitoes that were not exposed to any toxicant). Where multiple bottle bioassays were done on the same strain within a week (i.e., the same generation), enzyme content and activities were assessed on only one set of controls from that week. Because sample size for the controls varied between nine and thirty individuals, eight individual mosquitoes were used from each strain. To produce the enzyme source, mosquitoes were placed into 2.0 mL screw cap microcentrifuge tubes loaded with two 2.0 mm zirconium oxide beads and 400 μL of 100 mM sodium phosphate buffer (pH 7.4) with no chelating agents or protease inhibitors. Mosquitoes were homogenized with a bead mill and then centrifuged at 10,000 x g at 4°C for 10 min. The resulting supernatant was used as the enzyme source for all subsequent activity assays as well as for soluble protein determination.

The MFO content in samples was determined using the 3,3',5,5'-tetramethylbenzidine (TMB) method originally devised by [23]. This assay does not measure the activity of MFOs toward a model substrate. Instead, it measures the heme content in the sample, the majority of which is thought to belong to MFOs [24]. A 0.2% TMB solution was freshly prepared in methanol and then further diluted to 0.05% in 250 mM sodium acetate buffer (pH 5.0). A 20 μL volume of supernatant was added to a 96-well plate in duplicate followed by 200 μL of 0.05% TMB solution. The reaction was started by adding 25 μL of 3% $H_2O_2$ to each well and then incubating the plate at room temperature for 10 min inside a dark cabinet. Results were compared to a standard curve generated from cytochrome C using the same reagents and volumes as the samples ($R^2$ = 0.997). After the incubation period, wells were read at 620 nm in a BioTek Epoch 2 spectrophotometer (BioTek, Santa Clara, CA). Units of activity were thus reported as cytochrome equivalents.

For α- and β-CarE activity, 15 μL of supernatant was incubated with 135 μL of freshly prepared 0.3 mM α- or β-naphthyl acetate (final reaction concentration 0.27 mM in wells) in duplicate in a 96-well plate covered with a lid. Incubation lasted for 15 min at room temperature inside a dark cabinet. The reaction was stopped by adding 50 μL of freshly prepared 0.3% Fast Blue B in 5.0% sodium dodecyl sulphate solution. Color was allowed to develop for 5 min at room temperature in the dark cabinet. The α-naphthyl acetate (α-CarE) samples were read at 600 nm and the β-naphthyl acetate (β-CarE) samples were read at 550 nm. Standard curves of α- and β-naphthol were run in triplicate (both $R^2$ = 0.999) and used to quantify the enzymatic conversion of α- and β-naphthyl acetate to α- and β-naphthol, respectively.

For GST activity, 20 μL of supernatant was added in duplicate to wells of a 96-well plate. The substrate mixture consisted of 1 mL of 21 mM 1-chloro-2,4-dinitrobenzene (CDNB) in methanol added to 10 mL of freshly prepared 10 mM of reduced glutathione (GSH) in 100 mM sodium phosphate buffer (pH 6.5) immediately prior to being mixed with the supernatant. A pH of 6.5 was used for the substrate mixture to minimize auto conjugation of GSH with CDNB. 180 μL of substrate mixture was added to each well of supernatant for a final concentration of 0.9 mM CDNB and 0.86 mM GSH. The plate was read at 340 nm in 1 min intervals for 5 min. An experimentally derived extinction coefficient of 0.00580 μM$^{-1}$ cm$^{-1}$ was used that accounted for the path length through the 200 μL total volume in each well.

Total protein was measured with the Bradford method using bovine serum albumin as a standard ($R^2$ = 0.978) [25]. Units for all four enzyme assays were MFO content as μg cytochrome c equivalents/mg protein, and CarE and GST specific activities as nmol substrate/min/mg protein.

## Statistical analyses

All analyses were conducted in R version 4.1.1 [26]. Enzyme activities were compared by Kruskal-Wallis tests due to this test being robust to smaller sample sizes. When applicable, a Mann-

Whitney-U test was used for pairwise comparisons between the strains. Two analyses were done, one that included DPN1 and another that included DPN2. This was done to simplify the analyses because DPN1 and DPN2 are repeated measures of the same strain, thus not independent, and all other strains were independent. Each enzyme activity assay also was regressed against percent mortality at both 30 min and at 120 min and the slopes were used to determine an effect of enzyme activity on percent mortality. This was done by building general linear models, which were visually assessed for heteroscedasticity and normality of residuals.

For CDC bottle bioassays, a clustered Cox regression was generated on pairwise comparisons of Sumithrin® and Anvil® or Sumithrin® and AnvilTM at the 30 min diagnostic time point used by the CDC to assess resistance [19] using the 'survival' package [16]. The clustering effect was assigned to the multiple bottles for each of the treatments. Prior to statistical testing, the Cox models were assessed for proportionality using the 'cox.zph()' function. Models that violated the assumption of proportionality had a time-dependent coefficient added to them (i.e., an interaction term between time in minutes and treatment), and these hazard ratios are reported with a time factor change in hazard rate of mortality. For ties in times to death, the Efron approximation was used. A Wald test was used to test the null hypothesis that the beta coefficients = 0 at $\alpha = 0.05$ and results are reported as fold change in the hazard rate of mortality in the Anvil® or AnvilTM (both Sumithrin® + PBO) treatment compared to the Sumithrin® treatment. Bottle bioassays were analyzed only if at least three replicates of each treatment were done per location. Thus, analyses of AHB, WHE, and DPN were included.

## Results

### Bottle bioassays

The distribution of observed mortality over 30 min differed in several strains when comparing Sumithrin® technical (20 µg/bottle) against either Anvil® at 22.17 µg Sumithrin® and 22.17 µg PBO or an Anvil® tank mix (AnvilTM) equivalent with 11.09 µg Sumithrin® and 11.09 µg PBO in 1:1 mineral oil (Table 3). Mortality reached 100% in the Anvil® treatment by the diagnostic time of 30 min but was 70% against the technical Sumithrin®. There was a significant difference between Sumithrin® and Anvil® in AHB mosquitoes (Fig 3; HR = 3.85 (2.496–5.949 95% CI), P < 0.001). There was a 3.9-fold change in hazard rate of mortality when mosquitoes were exposed to the Anvil® treatment over the course of 30 min.

**Table 3. Percent mortality of five different strains of *Culex pipiens* complex when exposed to a diagnostic dose of Sumithrin® (20 µg/bottle), Anvil® (22.2 µg/bottle Sumithrin® + 22.2 µg/bottle PBO), or AnvilTM (11.1 µg/bottle Sumithrin® + 11.1 µg/bottle PBO cut 1:1 with mineral oil) from onset of treatment to the diagnostic time point (30 min) and the study end point (120 min).**

| | % Mortality[1] | | | | | |
| | After 30 min | | | After 120 min | | |
| Strain | Anvil® | AnvilTM | Sumithrin® | Anvil® | AnvilTM | Sumithrin® |
|--------|--------|---------|------------|--------|---------|------------|
| AHB | 100 | - | 70.0 | 100 | - | 95.7 |
| WHE | 100 | 88.8 | 51.4 | 100 | 97.2 | 86.5 |
| DPS | - | - | 10.5 | - | - | 68.4 |
| DPSB | 88.2 | 100 | 82.6 | 100 | 100 | 100 |
| DPN | 82.4 | - | 72.1 | 98.8 | - | 91.8 |
| DPN1 | 97.8 | 98.5 | 76.6 | 100 | 100 | 100 |
| DPN2 | - | - | 61.1 | - | - | 100 |

[1] CDC guidelines consider samples with a mortality at the diagnostic time of > 97% are susceptible, 96–90% are building resistance, and < 90% are resistant [19].

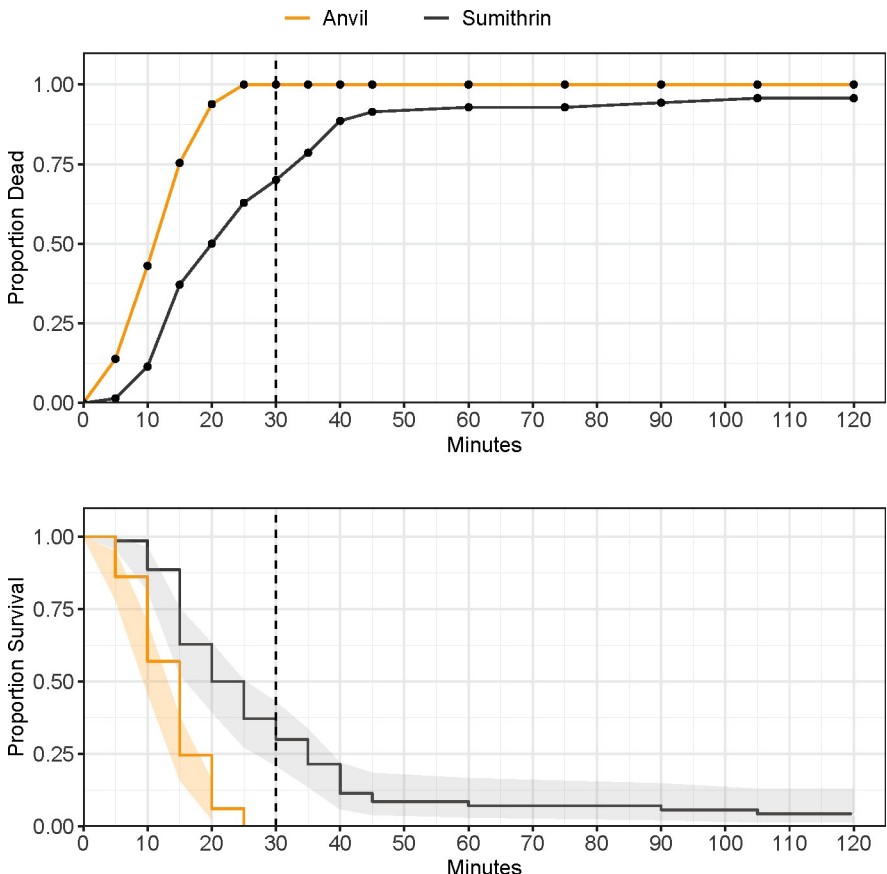

**Fig 3. Proportion mortality/survival of Arlington Heights catch basin (AHB) adult *Culex pipiens* complex in CDC bottle bioassays, with either Sumithrin or Anvil, after 120 minutes of continuous exposure.** Top panel: common mortality curve. Bottom panel: Kaplan-Meier survival curves and 95% confidence intervals (shaded area). Vertical dotted line at 30 minutes denotes the diagnostic time for Sumithrin®.

Mortality also reached 100% in Anvil® at the diagnostic time in WHE. The hazard ratio was not proportional over time by a factor of 0.91 (Fig 4; HR = 0.91 (0.841–0.992), P = 0.032), and there was a significant difference between Sumithrin® and Anvil® (HR = 36.11 (8.465–154.007 95% CI), P < 0.001). Thus, there was a 36.1-fold change in hazard risk of mortality in the Anvil® treatment at zero min that changed by 0.91 every minute (i.e., 36.1 x $0.91^n$, where n = minutes). At 30 min, the hazard rate of mortality in the Anvil® treatment was about a 2.1-fold compared to Sumithrin® alone. The hazard also was not proportional over time between Sumithrin® and AnvilTM at 30 min (HR = 0.91 (0.858–0.970), P = 0.003). The hazard ratio significantly differed between Sumithrin® and AnvilTM (HR = 17.33 (4.707–63.817), P < 0.001). There was a 17.3-fold change in hazard rate of mortality in the AnvilTM treatment starting at zero min and changed by 0.91 every minute for 30 min. By 30 min, the hazard rate of mortality was about double in the Anvil® treatment.

There was no difference in hazard rate of mortality between Sumithrin® and Anvil® in the DPN samples (Fig 5; HR = 1.26 (0.605–2.629), P = 0.537).

## Genotyping for knockdown resistance alleles

All five strains tested were positive to varying degrees for the 1014F allele (Table 4). The DPS strain had the highest percent homozygosity for the *kdr* genotype (FF), as well as the highest

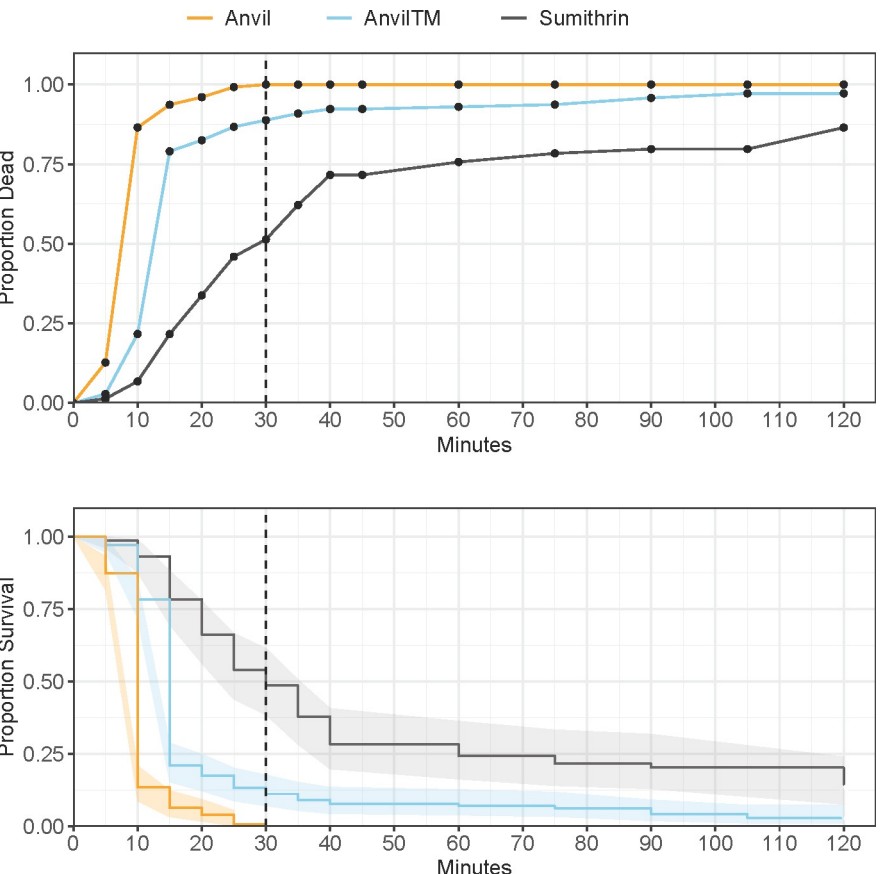

**Fig 4. Proportion mortality/survival of Wheeling (WHE) adult *Culex pipiens* complex in CDC bottle bioassays, with either Sumithrin, AnvilTM, or Anvil, after 120 minutes of continuous exposure.** Top panel: common mortality curve. Bottom panel: Kaplan-Meier survival curves and 95% confidence intervals (shaded area). Vertical dotted line at 30 minutes denotes the diagnostic time for Sumithrin.

frequency for the 1014F allele. No homozygotes for the susceptible genotype (LL) were identified among the tested DPS organisms. The DPN strain that was genotyped was from a collection earlier than those used in the bottle bioassay and enzyme activity assays and had the lowest percent homozygosity of the FF genotype, as well as the lowest 1014F allele frequency. Heterozygosity (genotype LF) was generally high compared to both homozygous genotypes in all five strains.

### Enzyme activity assays

The analysis was first done with all strains including DPN1 but not DPN2 and then all strains including DPN2 but not DPN1. Finally, DPN1 and DPN2 were compared. This was done because DPN1 and DPN2 were repeated measures of the DPN strain and thus not independent. Activity of GSTs were only significantly different when DPN2 was included in the group (Fig 6; $\chi^2$ = 10.59, df = 3, P = 0.014). AHB was significantly different from DPS (W = 54, P = 0.021) and DPN2 (W = 55, P = 0.015) and WHE was significantly different from DPN2 (W = 10, P = 0.021). No other enzyme activities or MFO content were significantly different among the strains. There also was no difference in any of the enzyme activities or MFO content between DPN1 and DPN2 (all P > 0.130). The coefficient of variation for α-CarE and β-CarE in the AHB strain were the two highest among all tested strains and enzyme activities (Table 5). There was no effect of enzyme activity or MFO content that explained the

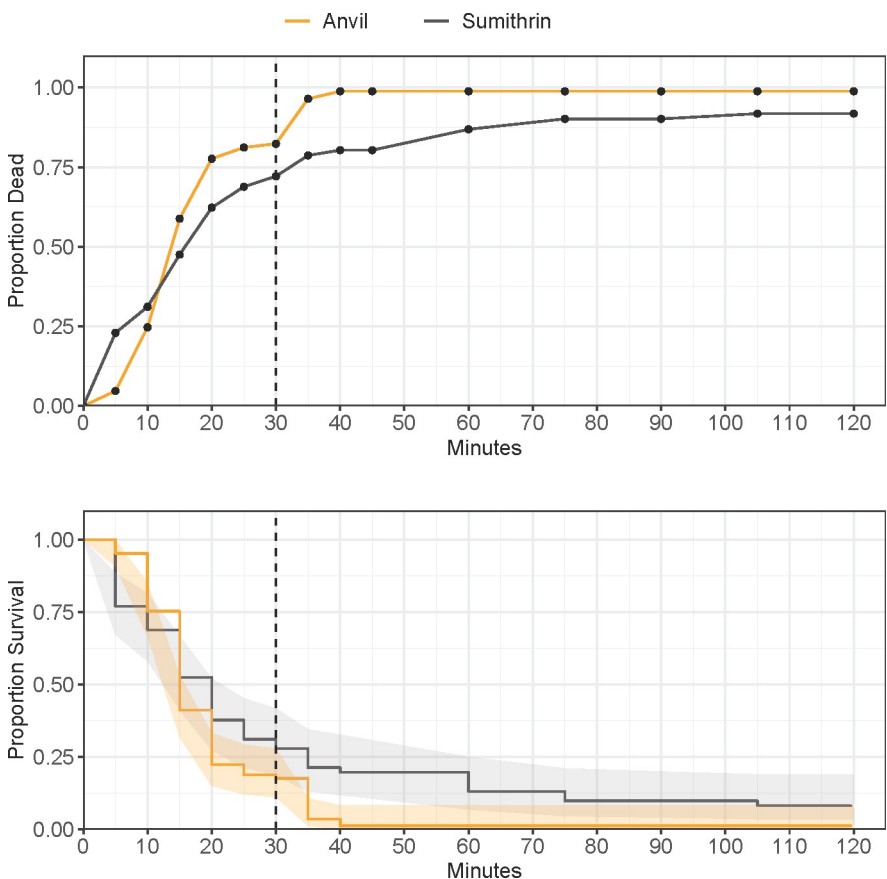

**Fig 5. Proportion mortality/survival of Des Plaines North (DPN) adult *Culex pipiens* complex in CDC bottle bioassays, with either Sumithrin or Anvil, after 120 minutes of continuous exposure.** Top panel: common mortality curve. Bottom panel: Kaplan-Meier survival curves and 95% confidence intervals (shaded area). Vertical dotted line at 30 minutes denotes the diagnostic time for Sumithrin.

differences in percent mortality at either 30 min or at 120 min (i.e., all regression slopes were statistically equal to zero).

## Discussion

According to CDC guidelines [19], all five strains in the present study may be resistant to Sumithrin®, and the formulated product Anvil® reclaimed efficacy in three out of five

**Table 4. Percent genotype and allele frequency of the leucine-to-phenylalanine (L1014F) knockdown resistance mutation in the voltage-sensitive sodium channel of adult female *Culex pipiens* complex from some Northwest suburbs of Chicago, Illinois.**

|  |  | Percent Genotype | | | Allele Frequency | |
| --- | --- | --- | --- | --- | --- | --- |
| Strain | n | FF | LF | LL | F | L |
| AHB | 44 | 25.0 | 68.2 | 6.8 | 0.59 | 0.41 |
| WHE | 46 | 41.3 | 54.3 | 4.3 | 0.68 | 0.32 |
| DPS | 37 | 43.2 | 56.8 | 0 | 0.72 | 0.28 |
| DPSB | 38 | 18.4 | 65.8 | 15.8 | 0.51 | 0.49 |
| DPN | 43 | 9.3 | 76.7 | 14.0 | 0.48 | 0.52 |

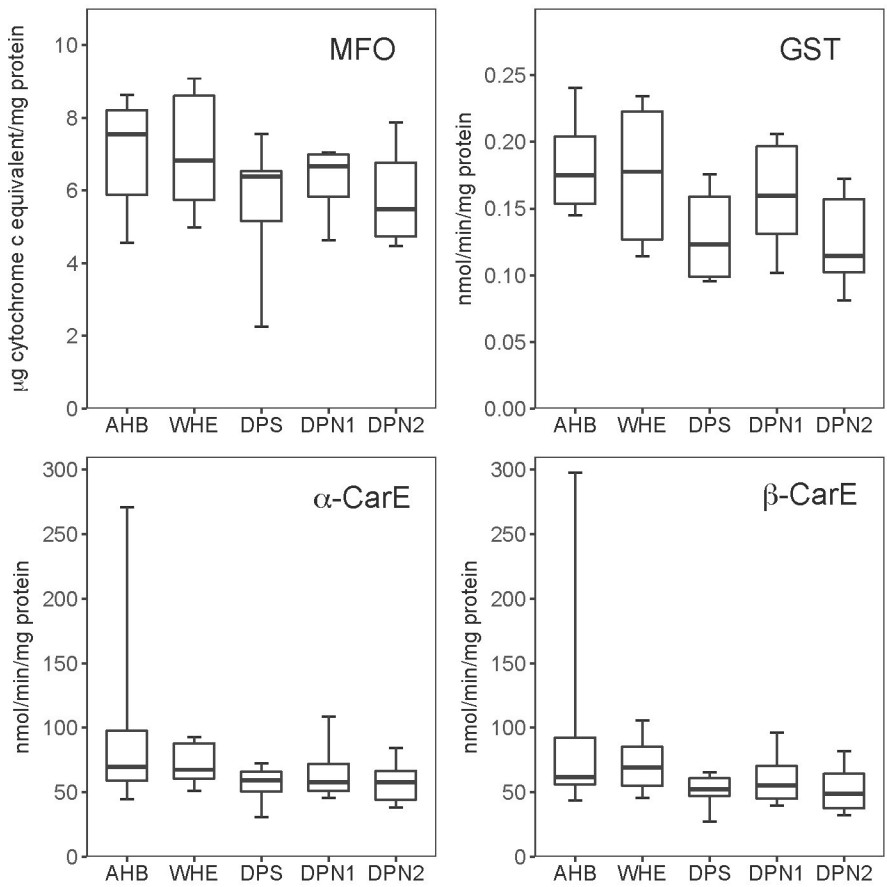

**Fig 6. Distributions of specific activities for mixed-function oxidase (MFO), glutathione *S*-transferase (GST), alpha-carboxylesterase (α-CarE), and beta-carboxylesterase (β-CarE) in female *Culex pipiens* complex from four different trapping locations in some Northwest suburbs of Chicago, Illinois.** The lower and upper hinge of boxes represent the 25th and 75th quartiles, respectively. The middle line represents the median. The whiskers represent the minimum and maximum range of the data.

strains. This is not surprising given the frequency of ULV applications in these areas during the time the mosquito collections were made. Previous studies have demonstrated resistance in both adult [27] and larval [28, 29] *Cx. pipiens* in the suburbs of Chicago. With historically high incidence of WNV in the Chicago area when compared to the rest of the upper Midwest and northeast US where *Culex pipiens* and *Culex restuans* are the main vector. [30], monitoring resistance status in *Cx. pipiens* is of public health importance. Knowing what mechanisms

**Table 5. Coefficient of variation for mixed-function oxidase (MFO), glutathione *S*-transferase (GST), alpha-carboxylesterase (α-CarE), and beta-carboxylesterase (β-CarE) specific activities in female *Culex pipiens* complex from four different trapping locations in some Northwest suburbs of Chicago, Illinois, USA.** Larger numbers indicate greater variability relative to their mean.

| Strain | MFO | GST | α-CarE | β-CarE |
|---|---|---|---|---|
| AHB | 23.6 | 19.6 | 76.5 | 87.3 |
| WHE | 22.7 | 28.0 | 23.1 | 29.0 |
| DPS | 30.6 | 25.6 | 25.6 | 23.7 |
| DPN1 | 13.9 | 24.6 | 34.2 | 35.2 |
| DPN2 | 22.2 | 28.4 | 28.3 | 33.1 |

these strains rely on to survive chemical control will help to inform mosquito abatement districts on application timing and product rotation.

Numerous resistance mechanisms have been implicated in *Cx. pipiens* (reviewed in [15]). Although *Culex* spp. are known to have a leucine-to-phenylalanine or leucine-to-serine single nucleotide polymorphism (SNP) at the 982$^{nd}$ amino acid in the para region of the voltage-sensitive sodium channel (structurally the same location as the L1014F/H mutation in house flies), their roles in resistance to pyrethroids is acknowledged but not well-resolved (e.g., [31, 32]). In the case of the leucine-to-serine mutation, there appears to be stronger resistance to DDT than to pyrethroids [33]. The house fly L1014F mutation is only known to confer moderate levels of resistance by itself [34] compared to bi- and tri-allelic *kdr* combinations [35]. In the case of the *Cx. pipiens* complex, hybridization rate may also play a role in expression of *kdr* [31]. In the present study, *kdr* alone does not appear to explain the relatively large differences in mortality to Sumithrin® at 30 and 120 min but does appear to have some involvement. The DPS strain had the lowest mortality (10.5%) at the diagnostic time of 30 min and had the greatest resistant phenylalanine (F) allele frequency, but WHE had similar allele frequency and resulted in over 40% greater mortality (51.4%) over 30 min. WHE had 100% mortality with Anvil® treatment, suggesting significant involvement of MFOs and/or CarEs.

Tests utilizing synergists in the CDC bottle bioassay can suggest enzymatic detoxification playing a role in resistance [36, 37], especially in the presence of high L982F *kdr* allele frequencies [38]. That we saw considerable restoration of susceptibility when these strains were tested with Anvil®, a Sumithrin® product that is synergized with PBO, is consistent with previous findings (e.g., [38]). Traditionally, PBO is thought to inhibit cytochrome P450s (aka MFOs), a key enzyme in many oxidative processes, including insecticide metabolism ([39]). But PBO also has been shown to inhibit carboxylesterases [40]. A correlative relationship with enzyme activity and resistance to deltamethrin, permethrin, and DDT in *Cx. quinquefasciatus*, a close relative to *Cx. pipiens*, has been previously shown [41]. Xu et al. [42] found over 1,000-fold difference in LC$_{50}$ in two resistant strains of *Cx. quinquefasciatus* that were selected with permethrin. PBO contributed significant reduction in LC$_{50}$ values, down to around 100-fold, of the resistant strains compared to the susceptible strain. Although methodology is different in the present study, the results of the bottle bioassays are consistent with PBO contributing significant restoration of Sumithrin® efficacy, suggesting that either cytochrome P450s and/or carboxylesterases play a role in the resistance phenotype among these strains. Similar restoration of susceptibility was recently demonstrated in populations of *Cx. quinquefasciatus* from Florida and California [36, 37].

In the case of the MFO activity assay, 3',5',5'-TMB is a substrate used to measure heme content in a sample, of which most is thought to be attributed to cytochrome P450s [23]. This assay is considered a surrogate assay and does not measure activity directly as in other types of activity assays, such as the CarE and GST assays in the present study. Although regressions of bottle bioassay mortality did not suggest a relationship based on MFO quantity, CarE, or GST activities in the control mosquitoes, it does not rule out altered enzyme activity toward Sumithrin®. It also does not rule out inducible overexpression of these enzymes, especially MFOs, which have been documented in resistant *Cx. quinquefasciatus* [43]. This is a plausible explanation for the discrepancy between enzyme activities and PBO synergism because induction usually happens shortly after initiation of exposure to a toxicant, and enzyme activities conducted in the present study were done on control mosquitoes (i.e., not exposed to Sumithrin®). Regardless of the dynamics of cytochrome P450 expression, PBO tends to reduce pyrethroid LD$_{50}$s even in lab-reared, fully susceptible *Culex* spp. (e.g., [37]), and *Musca domestica* (e.g., [44]).

Although limited in what they can tell us about enzymatic involvement in resistance, an application of enzyme activity assays that may be useful in describing the dynamics of the

enzymatic role in resistance is in an induction experiment. In an induction experiment, the researcher exposes the animals to a sublethal dose of the insecticide and collects individuals over a time series. The researcher then runs activity assays on the specimens and relates the activities back to toxicological response data. The assay in the present study only had the potential to detect constitutively expressed detoxification enzymes. The goal of a future study in *Cx. pipiens* complex adults from the Chicago area should include an induction experiment with Sumithrin®. That the synergized formulation of Sumithrin® and PBO in Anvil® restored susceptibility in multiple strains suggests that expression of MFOs and/or CarEs may be induced at high levels upon exposure to a toxicant like Sumithrin®, or perhaps even have a target site mutation that confers an advantage in processing Sumithrin®. Another explanation could be that the model substrates used in the present study simply do not interact with the isozymes that may play a role in *Cx. pipiens* complex resistance to Sumithrin®. This could also be true of any induction experiment undertaken in future studies.

We note that in the AHB strain, there was a high coefficient of variation in the CarE activities relative to the other strains. The inclusion of PBO with Sumithrin® appeared to provide satisfactory control in this strain at the time of data collection and should be considered in future regional control efforts. WHE had notable resistance to Sumithrin® but similarly the addition of PBO in Anvil® provided 100% mortality at the diagnostic time of 30 min. AHB and WHE had the greatest difference in mortality at 30 min diagnostic period between Sumithrin® and Anvil® (30% and 48.6%, respectively). The DPS strain showed the highest resistance according to CDC diagnostic dose and time for Sumithrin®. DPS also showed the highest F allele frequency and the highest percentage of resistant FF homozygotes. DPS was not screened with Anvil® and there was no indication of enzymatic involvement based on the activity assays. A follow up study should focus on the DPS strain and include an Anvil® bottle bioassay to measure an effect, if any, that MFOs and/or CarEs have on its resistance. Interestingly, neither DPSB nor two of the DPN resamples reverted to 100% mortality when treated with Anvil®. By the second resampling (DPN2), resistance to Sumithrin® increased by 11%.

Examined from the perspective of hazard ratios, in the case of AHB and WHE, we saw that treatment with Anvil® resulted in a 3.9-fold and 2.1-fold change in the risk of mortality at 30 min compared to Sumithrin®. In DPN, the raw bottle bioassay mortality data shows only a small increase in mortality in the Anvil® treatment, which statistically conferred no greater risk of mortality than the Sumithrin® treatment. With the inability to calculate informative numbers such as resistance and synergist ratios (e.g., [45, 46]) using diagnostic doses in bottle bioassays, hazard ratios from Cox regression offer the potential to make ratio-based quantitative comparisons among treatments. In the present study, the choice to analyze bottle bioassay data using Cox regression was made after bottle bioassays had already been run and some potentially important factors such as sex and genotypes were not collected. The inclusion of these types of factors would help to explain results like these in future studies. We recommend the use of clustered Cox regression with time-dependent covariates when describing how much risk multiple variables contribute to the rate of mortality across a given diagnostic time period. Typical analyses of bottle bioassays, including binomial generalized linear models and repeated measures ANOVA, violate several important assumptions of these statistical tests, including the independence of observations [47], which seldom allow for the correct grouping variables to account for this fact. This can be accounted for using a clustering effect in Cox regression, which is akin to a random factor in a mixed model. Quite often, studies that utilize Kaplan-Meier survival analysis are not assessed for the assumption of proportional hazards, a key assumption that must be met for model estimates to be deemed accurate [16]. This is evident by the often-seen crossing of survival curves in a figure (e.g., Fig 4). We have provided code for readers to use on future analyses of bottle bioassay data using Cox regression. Cox

regression is a benefit to the interpretation of bottle bioassay data because it allows multiple factors to be assessed simultaneously across numerous time points, not just a single endpoint. We liken these two statistical comparisons to taking a picture (end-point analysis) versus watching a movie (Cox regression analysis). For instance, in Fig 3 (WHB) mortality in the Anvil® and AnvilTM treatments were in excess of 75% within the first 10 and 15 min, respectively, at which point an inflection in the rate of mortality can be seen. With downstream molecular and biochemical analyses, this point of inflection could be characterized, with Cox regression as the statistical method to infer risk of mortality due to those types of factors. Future studies on *Cx. pipiens* resistance in field strains, where topical application of pesticides is not feasible, should incorporate this type of analysis to make clearer distinctions among the numerous resistance mechanisms and their relative impact on operational success.

## Supporting information

**S1 File. R-script for Cox regression survival analysis of bioassay data.**
(PDF)

## Acknowledgments

The authors thank Skyler Finucane, Natalia Szklaruk, Jim Downing, Colin Murphy, and Jack Ponterelli for assistance with mosquito rearing and bottle bioassays.

Its contents are solely the responsibility of the authors and do not necessarily represent the official views of the Centers for Disease Control and Prevention or the Department of Health and Human Services or the USDA. Mention of trade names or commercial products in this publication is solely for the purpose of providing specific information and does not imply recommendation or endorsement by the U.S. Department of Agriculture. USDA is an equal opportunity provider and employer.

## Author Contributions

**Data curation:** Kristina Lopez, Patrick Irwin.

**Formal analysis:** Edwin R. Burgess, IV, Collin P. Jaeger.

**Investigation:** Edwin R. Burgess, IV, Kristina Lopez, Patrick Irwin, Alden S. Estep.

**Methodology:** Edwin R. Burgess, IV, Collin P. Jaeger, Alden S. Estep.

**Project administration:** Edwin R. Burgess, IV, Patrick Irwin.

**Supervision:** Patrick Irwin, Alden S. Estep.

**Writing – original draft:** Edwin R. Burgess, IV.

**Writing – review & editing:** Edwin R. Burgess, IV, Kristina Lopez, Patrick Irwin, Collin P. Jaeger, Alden S. Estep.

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
