## [Decision Letter · Decision Letter 0]

17 May 2022

PONE-D-22-11736

Assessing pyrethroid resistance status in the Culex pipiens complex (Diptera: Culicidae) from the northwest suburbs of Chicago, Illinois using Cox regression of bottle bioassays and other detection tools

PLOS ONE

Dear Dr. Estep,

Thank you for submitting your manuscript to PLOS ONE. After careful consideration, we feel that it has merit but does not fully meet PLOS ONE’s publication criteria as it currently stands. Therefore, we invite you to submit a revised version of the manuscript that addresses the points raised during the review process.

We look forward to receiving your revised manuscript.

Kind regards,

Ahmed Ibrahim Hasaballah

Academic Editor

PLOS ONE

Journal Requirements:

Reviewers' comments:

Reviewer's Responses to Questions

Comments to the Author

1. Is the manuscript technically sound, and do the data support the conclusions?

Reviewer #1: Yes

Reviewer #2: Yes

2. Has the statistical analysis been performed appropriately and rigorously?

Reviewer #1: Yes

Reviewer #2: Yes

3. Have the authors made all data underlying the findings in their manuscript fully available?

Reviewer #1: Yes

Reviewer #2: No

4. Is the manuscript presented in an intelligible fashion and written in standard English?

Reviewer #1: Yes

Reviewer #2: Yes

5. Review Comments to the Author

Reviewer #1: I've made comments on a copy of the manuscript and will upload that.

Overall the paper is very well written and contributes to what is known about resistance in an important vector of West Nile

virus. I made a few comments and suggestions. Nice work.

Reviewer #2: Good study reporting the situation of pyrethroid resistance in some Culex pipiens populations in Chicago Illinois I have few remarks

Methods

In the section “Mosquito sources and pyrethroid exposure histories” the authors should add information on common aquatic habitats for Culex larvae in the different study sites.

Line 123 to 128 “Sprays were conducted once a week for 5 weeks in Wheeling, Arlington Heights North, and Des Plaines South in 2019 and 2020, starting in July and ending in August. In 2019, Zenivex® E20 (20% etofenprox) in a 1:1 mix with mineral oil (10% etofenprox) was used and in 2020, Anvil® 10+10 was sprayed at 0.0036 Lb per acre of active ingredient and piperonyl butoxide (PBO). From 2013 – 2018 Northwest Mosquito Abatement District averaged 1 spray event (etofenprox) per year in these areas.”

What area is sprayed? is it houses? Water collection? What??? Please add information

Results

The authors say they did mosquito identification in the methodology but no information on the identity of species recorded by sites is presented.

The authors also say haven used both male and females for bioassay but no information on the susceptibility of each group to insecticides is provided. I will propose that they add information on males and females susceptibility to insecticide. This data could improve comprehension and interpretation of their findings.

Genotyping for knockdown resistance alleles

The authors should include details of mosquito processed were these mosquitoes survivors to bioassays? were they dead? or mosquitoes not exposed?

Enzyme activity assays

The authors should add the precision whether the expression profile reported is for resistant samples or for samples not exposed to insecticide.

6. PLOS authors have the option to publish the peer review history of their article (what does this mean?). If published, this will include your full peer review and any attached files.

Do you want your identity to be public for this peer review? For information about this choice, including consent withdrawal, please see our Privacy Policy.

Reviewer #1: Yes: Linda Kothera

Reviewer #2: No

---

## [Author Response · Author response to Decision Letter 0]

6 Jun 2022

A thorough response to both the Editor and Reviewer comments are in the Response to Reviewers

---

## [Editor Report · Decision Letter 1]

10 Jun 2022

Assessing pyrethroid resistance status in the *Culex pipiens* complex (Diptera: Culicidae) from the northwest suburbs of Chicago, Illinois using Cox regression of bottle bioassays and other detection tools

*PONE-D-22-11736R1*

*Dear Dr. Estep,*

*We’re pleased to inform you that your manuscript has been judged scientifically suitable for publication and will be formally accepted for publication once it meets all outstanding technical requirements.*

*Within one week, you’ll receive an e-mail detailing the required amendments. When these have been addressed, you’ll receive a formal acceptance letter and your manuscript will be scheduled for publication.*

*An invoice for payment will follow shortly after the formal acceptance. To ensure an efficient process, please log into Editorial Manager at http://www.editorialmanager.com/pone/, click the 'Update My Information' link at the top of the page, and double check that your user information is up-to-date. If you have any billing related questions, please contact our Author Billing department directly at authorbilling@plos.org.*

*If your institution or institutions have a press office, please notify them about your upcoming paper to help maximize its impact. If they’ll be preparing press materials, please inform our press team as soon as possible -- no later than 48 hours after receiving the formal acceptance. Your manuscript will remain under strict press embargo until 2 pm Eastern Time on the date of publication. For more information, please contact onepress@plos.org.*

*Kind regards,*

*Ahmed Ibrahim Hasaballah*

*Academic Editor*

*PLOS ONE*

---

## [Editor Report · Acceptance letter]

20 Jun 2022

PONE-D-22-11736R1 

Assessing pyrethroid resistance status in the *Culex pipiens* complex (Diptera: Culicidae) from the northwest suburbs of Chicago, Illinois using Cox regression of bottle bioassays and other detection tools 

Dear Dr. Estep:

I'm pleased to inform you that your manuscript has been deemed suitable for publication in PLOS ONE. Congratulations! Your manuscript is now with our production department. 

Kind regards, 

on behalf of

Dr. Ahmed Ibrahim Hasaballah 

Academic Editor

PLOS ONE